# Individual consistency in the behaviors of newly-settled reef fish

James R. White[1,2], Mark G. Meekan[3] and Mark I. McCormick[1,2]

[1] College of Marine and Environmental Sciences, James Cook University, Townsville, Queensland, Australia
[2] ARC Centre of Excellence for Coral Reef Studies, James Cook University, Townsville, Queensland, Australia
[3] Australian Institute of Marine Science, University of Western Australia, Crawley, Western Australia, Australia

## ABSTRACT

Flexibility in behavior is advantageous for organisms that transition between stages of a complex life history. However, various constraints can set limits on plasticity, giving rise to the existence of personalities that have associated costs and benefits. Here, we document a field and laboratory experiment that examines the consistency of measures of boldness, activity, and aggressive behavior in the young of a tropical reef fish, *Pomacentrus amboinensis* (Pomacentridae) immediately following their transition between pelagic larval and benthic juvenile habitats. Newly-settled fish were observed in aquaria and in the field on replicated patches of natural habitat cleared of resident fishes. Seven behavioral traits representing aspects of boldness, activity and aggression were monitored directly and via video camera over short (minutes), medium (hours), and long (3 days) time scales. With the exception of aggression, these behaviors were found to be moderately or highly consistent over all time scales in both laboratory and field settings, implying that these fish show stable personalities within various settings. Our study is the first to examine the temporal constancy of behaviors in both field and laboratory settings in over various time scales at a critically important phase during the life cycle of a reef fish.

## INTRODUCTION

There has been considerable interest in, and evidence for, consistent patterns in the behaviors of individual animals within a species over the last decade (*Dall, Houston & McNamara, 2004*; *Sih, Bell & Johnson, 2004*; *Sih et al., 2004*; *Dingemanse & Réale, 2005*; *Bell, 2007*; *Réale et al., 2007*; *Smith & Blumstein, 2008*; *Bell, Hankison & Laskowski, 2009*). Differences in the amount of aggressive, exploratory and bold behaviors among individuals have been shown to be widespread and heritable (*Boake, 1994*; *Stirling, Reale & Roff, 2002*; *Kolliker, 2005*; *Van Oers et al., 2005*; *Réale et al., 2007*) across a diverse array of taxa (*Dingemanse & Réale, 2005*; *Smith & Blumstein, 2008*) and to influence survival (*Downes, 2002*; *Dingemanse et al., 2004*), reproductive success (*Both et al., 2005*; *Sih & Watters, 2005*; *Pruitt & Ferrari, 2011*), resource acquisition (*Webster, Ward & Hart, 2009*) and growth.

Corresponding author
James R. White,
james.ryan.white@gmail.com

Adopting a certain behavioral phenotype can have both costs and benefits, for example, highly aggressive female fishing spiders (*Dolomedes triton*) are more successful at acquiring food in a competitive environment, but this aggression can be detrimental in another context such as when it leads to precopulatory sexual cannibalism (*Johnson & Sih, 2005*). Thus, consistent patterns in behavior among individuals can lead to trade-offs in aspects of fitness, which can ultimately influence population dynamics, community structure, and species diversity (*Pruitt, Grinsted & Settepani, 2013*; *Mittelbach et al., 2014*).

Variation in consistent behavioral patterns among individuals have been variously (and interchangeably) termed 'behavioral syndromes,' 'temperament,' 'personality,' and 'coping styles' (*Dall, Houston & McNamara, 2004*; *Réale et al., 2007*; *Dingemanse et al., 2010*; *Sih et al., 2012*), although some authors have argued for a more restrictive use of terminology (*Bell, Hankison & Laskowski, 2009*; *Garamszegi & Herczeg, 2012*). Here, we adopt the definitions of *Garamszegi & Herczeg (2012)*, where consistency in single behaviors (e.g., individuals that display repeatedly higher or lower levels of boldness, exploration, or aggression than others in the population) are described as displaying 'personality,' and consistency in the relationship between two or more functionally different behaviors within the same individual is defined as a 'behavioral syndrome.' For example, a behavioral syndrome is evident in the correlation between boldness and aggression documented within individual sticklebacks (*Gasterosteus aculeatus*) (*Bell, 2005*) and funnel-web spiders (*Agelenopsis aperta*) (*Riechert & Hedrick, 1993*).

Although the ability to alter behavior to suit changing environmental conditions is likely to be advantageous (*Kelley, Phillips & Evans, 2013*), behavior is not infinitely plastic (*DeWitt, Sih & Wilson, 1998*). If a single optimal behavioral phenotype existed, natural selection should reduce genotypic variation over generations (*Réale et al., 2007*). Because behavioral phenotypes show heritable variation not eroded by selection (*Penke, Denissen & Miller, 2007*; *Réale et al., 2007*), different behavioral strategies are likely to have different associated costs and benefits (*Kelley, Phillips & Evans, 2013*). For example, larger, bolder and faster-growing phenotypes of rainbow trout (*Oncorhynchus mykiss*) are more likely to be captured by fishing gears (*Biro & Post, 2008*).

Estimating the consistency of a behavioral trait is necessary for measuring the repeatable characteristics of a focal organism, quantifying trait plasticity and determining trait heritability (*Nakagawa et al., 2007*). Historically, personality studies using a single assay were common, but it has been recently suggested that repeated tests are essential for any personality study (*Réale et al., 2007*) and the strength of behavioral syndromes are likely underestimated when based upon single assays of varying traits (*Adolph & Hardin, 2007*; *Beckmann & Biro, 2013*; *White et al., 2013*; *White, McCormick & Meekan, 2013*).

Clearly, there is a need to determine the consistency of behaviors before examinations of personality, behavioral syndromes and associated trade-offs of alternative behavioral strategies can be attempted. Here, we examine evidence for personalities in a juvenile tropical reef fish, the Ambon damselfish (*Pomacentrus amboinensis*), by establishing the consistency of commonly-used field and laboratory assays of activity, aggression and boldness over time scales ranging from minutes to days following settlement. Similar

to many reef fishes, young of this species can be collected at the end of their larval phase immediately prior to settlement on the reef, when they are naïve to reef-based predators and behaviors learned after settlement (*Meekan et al., 2010*). In this immediate post-settlement phase of their life cycle, reef fishes typically experience very high mortality (*Almany & Webster, 2006*), with rates within the first 48 h of benthic life averaging 57% (*Doherty et al., 2004*; *Almany & Webster, 2006*). Because experience can influence behavioral phenotypes (*Budaev, 1997*; *Bell & Sih, 2007*; *Dingemanse et al., 2009*), the use of naïve study organisms allows us to control for variation and consistency in behavior associated with experience and to examine ecologically important behavioral traits at a critical ontogenetic boundary (*McCormick & Meekan, 2010*; *Poulos & McCormick, 2014*). Because field measurements are made directly by an observer on SCUBA (where visual and auditory presence is not easily concealed), we tested for an effect of observer presence by comparing observed behaviors to those recorded by video-camera. Specifically, we aimed to determine if juvenile damselfish behaviors were: (1) significantly altered by observer presence; (2) consistent over various time scales (minutes, hours, days) relevant to their major mortality bottleneck (first 48 h following settlement); (3) consistent in an aquarium setting; and (4) correlated between field and lab-based measurements. Based on our anecdotal previous experience with this system and study species, we predicted all behaviors to be moderately consistent in the field and laboratory.

## METHODS

### Ethics statement

Fish collection locations/activities and handling protocols were approved by the Great Barrier Reef Marine Park Authority (Permit Number: G10/33784.1) and JCU Animal Ethics Committee (Permit Number: A1720). All efforts were made to minimize animal handling and stress.

### Study site and species

This study was conducted on the shallow reef (2–4 m depth) offshore from the Lizard Island Research Station (14°40′S, 145°28′E) on the northern Great Barrier Reef, Australia. Our study species, the Ambon damsel, *P. amboinensis*, is common on Indo-Pacific coral reefs (*Beukers & Jones, 1998*). After approximately 20 days as pelagic larvae and at about 11 mm standard length (*Wellington & Victor, 1989*), young fish settle from the plankton at night to reefs (*Pitcher, 1988*). These fish preferentially choose to settle on live coral (*McCormick & Weaver, 2012*) and settlement occurs predominantly between October and January around the time of the new moon (*Meekan, Milicich & Doherty, 1993*). Newly settled fish are found as solitary individuals associated with conspecific adults and sub-adults (*McCormick & Makey, 1997*). *P. amboinensis* has a relatively small home range (*Brunton & Booth, 2003*), moving only small distances (<1 m) during the first few months after settlement (*McCormick & Makey, 1997*). Due to its high abundance, small size, rapid development, and sedentary nature, *P. amboinensis* is an ideal model organism for field and laboratory based behavioral studies (*Meekan et al., 2010*).

## Experimental design
### Collection
We collected newly-metamorphosed juveniles of *P. amboinensis* (*McCormick & Makey, 1997*) using moored light traps (see small light trap of Fig. 1 in *Meekan et al., 2001* for design) during the October recruitment pulse. Different cohorts of fish were used for the different experiments. Traps were anchored approximately 100 m from the nearest reef in ~10 m of water at dusk and left overnight. Catches were emptied from the traps the next morning between 05:30 and 07:00 h. All fish collected from the traps were transported to the laboratory where *P. amboinensis* was separated from all other species and maintained in a 25 L aquarium (at densities <100 individuals/25 L) of aerated seawater for 24 h to acclimatize to local conditions and reduce handling stress before experiments began. Fish were fed *Artemia* nauplii twice daily while in captivity. For field experiments, each acclimated *P. amboinensis* was transported to the field in individually-labeled clip-seal plastic bag. After final observations, study organisms were released unharmed on nearby natural habitat.

## Observational protocol
### Behavioral consistency in the field
All behavioral observations were made on individual fish in the field or aquaria in the laboratory using separate groups of fish for each assessment. Each *P. amboinensis* was placed into a labeled 2 L clip-seal plastic bag containing aerated seawater and transported to the field. Divers released an individual fish onto a small patch reef (30 × 30 × 30 cm) constructed from live and dead pieces of the bushy hard coral *Pocillopora damicornis* on the shallow (3–4 m water depth) sand flat. *P. amboinensis* recruits occur naturally in this habitat. Reefs were deployed in a single row, approximately 3 m apart, parallel to and 5 m from the nearest area of natural reef. Means and ranges of temperatures did not vary among reefs or among aquaria (M McCormick, 2009 & 2012, unpublished data) and care was taken in reef construction to ensure that patch reefs had only very minor differences in habitat structure. Previous studies have shown that such minor variation in topographic complexity of patch reefs has no effect on behavior of young fish (*McCormick & Meekan, 2010*; *Meekan et al., 2010*). Before introduction of the study fish, patch reefs were cleared of any resident fishes using hand nets. These were released on nearby natural reef far enough away to prevent their return (approx. 10 m). Individual study fish were then released onto their respective patch reefs and the first behavioral variable (latency to enter a novel environment; see description below) was recorded. Immediately afterwards, small wire cages (about 30 × 30 × 30 cm, 12 mm mesh size) were placed over the patch to allow the fish to acclimate to the new surroundings while being protected from predation. Cages were left a minimum of 20 min and carefully removed immediately before observations. Following established protocols outlined below (*McCormick & Meekan, 2010*; *Meekan et al., 2010*; *White et al., 2013*), divers conducted observations from at least 1 m away (with the aid of a 2 x magnifying glass) to avoid any effects that may have been caused by the proximity of the observer to the target fish.

*Short term consistency.* Three behavioral measures of activity were recorded simultaneously over a 3 min observation interval for each fish ($n = 18$) during October 2009: bite rate (number of feeding strikes towards objects floating in the water column); distance ventured (DV; the maximum distance in centimeters fish moved away from their patch reef); and height on the reef (categorized as a cumulative proportion of the time spent at varying heights over the 3 min observation period, with the top of the patch taken as height of 1, middle of the patch a height of 0.5, and bottom a height of 0). Relative height on the patch was summarized as a cumulative proportion of the time spent at varying heights over the 3 min observation period, calculated from the sum of the proportions multiplied by the height categories (0, 0.5, or 1). Following the 3 min interval, a 30 × 30 cm acrylic mirror (mounted on a 1 m PVC pole) was gently placed 10 cm in front of the focal fish. After a 1 min acclimation period, two scores of aggression were recorded as latency until first strike ('attack latency') and 'mirror strike rate' (combined number of strikes or tail whips) made toward their reflection over 3 min was recorded (*Gerlai, 2003*; *Marks et al., 2005*). To examine the level of behavioral consistency over a 2 h period, the entire suite of behavioral assays were repeated three times with 30 min between observations over a single day.

*Consistency over multiple days in field.* A separate sample of fish ($n = 21$) was used to assess behavior over multiple days in October 2012. Observations were made 3 times each day (at 9:00, 12:00, 16:00 h) for each of 3 days giving a total of 9 repeated observations per individual. During each observation, activity (bite rates, distance ventured (DV), and height) was recorded as described earlier.

*Observer vs. video.* To assess if there were any effects of observer presence, behaviors were recorded with a GoPro Hero 2™ high definition video camera (720p resolution) and compared against observer scores ($n = 29$) using fish collected in October 2012. The camera was placed 30 cm from focal fish and left to record for 10 min. The first observation was a 3 min period of the behaviors recorded by the observer (1 m away) and camera simultaneously. The second observation was the last recorded 3 min of video (without an observer present). For analysis, this provided three data sets for every fish: 'observer,' the 'simultaneous video' recorded at the same time as the direct observation, and the 'video' recording without observer presence. Because of the difficulty in discerning distance in the video, only bite rates and height (see below) were recorded and observations in which fish moved out of view of the camera for more than 20 s in total were discarded. Although the recording of observations (observer, simultaneous video and video) in the same order could have potentially introduced a habituation effect, we followed this protocol because it minimized disturbance to fish.

### Behavioral consistency in the laboratory

*Short term consistency.* Individual fish ($n = 10$) were assessed for boldness during the 2012 field season using a variation of a common test, latency to emerge from a shelter (*Budaev, 1997*; *Fraser et al., 2001*; *Brown, Jones & Braithwaite, 2005*; *Chang et al., 2012*). Each fish was gently transferred via hand net into an opaque ~162 cm³ plastic holding

chamber within an aquaria (13 L, 20 cm water depth) that also contained a small refuge of live *Pocillopora damicornis* at the opposite end and allowed to acclimatize for 30 min. The holding chamber was believed to be of adequate size because the fish displayed no apparent signs of confinement stress. The sides of each aquarium were blacked out with plastic sheeting to isolate them from neighboring tanks. After acclimation, observers standing behind a blind (black plastic sheeting) gently revealed the opening to the holding chamber. Time to emerge ('latency to emerge': defined as more than half of the body length outside of the holding chamber), was recorded for each fish with a cut-off time for the observation of 180 s. Location (categorized as a cumulative proportion of the time spent in various sections of the aquaria, with the third of the aquaria with the chamber given a value of 1, middle third of the aquaria a value of 0.5, and the third with coral refuge a value of 0) was recorded in the 5 min following emergence. A location score was calculated from the sum of the proportions multiplied by the location categories. Here, a lower location score represents a bolder fish. To get to the coral refuge they must exit the chamber and swim across the length of the aquaria, while a shyer fish would not risk leaving the chamber. Aggression was tested by gently placing an acrylic mirror ($30 \times 15$ cm) upright on the back wall of the aquaria, with the aquaria orientated lengthwise to the observer. Traits of aggression were measured in the same manner as in the field, as outlined earlier. Water flow was shut off during the acclimation period and behavioral observations to reduce auditory disturbance, but a gentle air flow through air stones was maintained to ensure adequate dissolved oxygen levels. Fish were fasted for 12 h before trials and fed *Artemia* upon completion to prevent varying hunger levels of individual fish potentially confounding behaviors. Assays were repeated 3 times over a 2 h period throughout a single day.

*Field vs. laboratory.* One sample of fish ($n = 32$) was compared across field and aquaria settings in 2012. In the morning (9:00) *P. amboinensis* within 2 d of capture by light traps were assessed for boldness (latency to emerge and location) and aggression (attack latency and strikes) in aquaria as described above. Later that afternoon (13:00) they were assessed for release latency, bite rate, distance ventured, height, and aggression (attack latency and mirror strike rate) in the field as described earlier. After resident fish were cleared from the patch reefs, each damselfish was carefully released from the plastic bag onto the sand 10 cm from the patch reef. Latency to emerge was the amount of time it took for the fish to move onto refuge of the patch reef and was timed from the moment the fish exited the bag, to the instant it reached the edge of the reef shelter.

## Data analysis

For all fish (total $n = 110$), consistency was calculated with a repeatability score (R), defined as the intra-class correlation coefficient (ICC), representing the fraction of total variation in a set of measurements attributable to the variance among individuals (*Wolak, Fairbairn & Paulsen, 2012*). R was calculated by constructing a general linear mixed model with individual (ID) included as a random factor in a one-way analysis of variance (ANOVA) model, with the transformed behavioral score as the dependent variable. All scores were $\log_{10}(x + 1)$ transformed to meet the assumption of normality and

linearity. The ratio of variance explained by among-individual variance to total variance calculated from an ANOVA represents a common measure of repeatability of each behavior (*Lessells & Boag, 1987*). Confidence intervals (CI) around each repeatability estimate were calculated using the exact confidence limit equation in *Searle (1971)*, which has been shown to be precise for this type of dataset (*Donner & Wells, 1986*; *Wolak, Fairbairn & Paulsen, 2012*). The R value indicates the strength of repeatability and ranges from 0 to 1, with values approaching one indicating high repeatability (*Briffa & Greenaway, 2011*). The *p*-value associated with the ANOVA is then used to determine if repeatability is significantly greater than zero (*Lessells & Boag, 1987*).

Relationships between behavioral traits observed in the field and aquaria were analyzed using Pearson's product-moment correlation. All scores were $\log_{10}(x + 1)$ transformed to improve normality. Statistical analysis used SPSS version 20.0 (SPSS Inc., Chicago, Illinois, USA).

## RESULTS

### Short term consistency in the field

In the field, activity measurements (bite rate, DV, and reef height) were highly repeatable, with repeatability scores between 0.52 and 0.69 ($n = 18$, Table 1). The aggression measures (attack latency and mirror strike rate) decreased over time and were not significantly repeatable. By the third observation, fish did not respond to their reflection aggressively at all, suggesting that they became habituated to the mirror.

### Consistency over multiple days in field

Fish sampled three times a day for 3 days also displayed activity (bite rate, DV, and height) behaviors that were moderately to highly consistent ($n = 21$, $R = 0.33$–$0.77$; Table 1).

### Observer vs. video

Observer and simultaneously collected video data were very consistent ($n = 29$, $R = 0.46$ bite rate, 0.76 reef height: Table 1), as were the two video observations ($n = 29$, $R = 0.69$ bite rate, 0.89 reef height; Table 1).

### Short-term consistency in the laboratory

The measure of boldness (i.e., latency to emerge) and location after emergence were moderately consistent ($n = 10$, $R = 0.38$ and 0.54 respectively; Table 1).

### Field vs. laboratory

There were only two significant correlations between field and laboratory-based measurements of behavior, with a moderate positive correlation between latency to emerge values in the field and the lab ($n = 32$, $r = 0.35$, $p = 0.049$; Table 2) and between field and lab measures of aggression latency ($n = 32$, $r = -0.385$, $p = 0.030$; Table 2). The other variables (i.e. measures of location and aggression) showed no evidence of consistency between laboratory and field measurements, suggesting that the behaviors are context dependent and laboratory measures have little relevance to field studies.

**Table 1 Repeatability (R) values with 95% confidence intervals (CI) for various measures of boldness and activity for juvenile Ambon Damselfish (*Pomacentrus amboinensis*).** For the observer vs. video section, the human observation is labeled 'observer,' the simultaneous video camera recording 'simultaneous video,' and the independent video recording 'video.'

| Trait | R | $p$ | R CI low | R CI high |
|---|---|---|---|---|
| **Field** | | | | |
| **Short term consistency** ($n = 18$) | | | | |
| Bite rate | 0.64 | <0.001 | 0.39 | 0.83 |
| Distance ventured[a] | 0.69 | <0.001 | 0.46 | 0.86 |
| Reef height | 0.52 | <0.001 | 0.24 | 0.76 |
| Aggression latency | 0.20 | NS | 0.07 | 0.52 |
| Aggression strikes | 0.20 | NS | 0.07 | 0.52 |
| **Multiple days** ($n = 21$) | | | | |
| Bite rate[a] | 0.77 | <0.001 | 0.64 | 0.88 |
| Distance ventured | 0.62 | <0.001 | 0.45 | 0.79 |
| Reef height | 0.33 | <0.001 | 0.16 | 0.55 |
| **Observer vs. video** ($n = 29$) | | | | |
| *Observer vs. simultaneous video* | | | | |
| Bite rate | 0.46 | 0.005 | 0.13 | 0.71 |
| Reef height | 0.76 | <0.001 | 0.56 | 0.88 |
| *Simultaneous video vs. video* | | | | |
| Bite rate | 0.69 | <0.001 | 0.45 | 0.84 |
| Reef height[a] | 0.89 | <0.001 | 0.79 | 0.95 |
| **Laboratory** | | | | |
| **Short term consistency** ($n = 10$) | | | | |
| Latency to emerge[a] | 0.38 | 0.026 | −0.004 | 0.76 |
| Location | 0.54 | 0.003 | 0.16 | 0.84 |

**Notes.**
[a] Individual reaction norm graph available in supplementary materials.

**Table 2 Pearson's product-moment correlations between field and laboratory measures of boldness and aggression for juvenile Ambon damselfish (*Pomacentrus amboinensis*).** All data ($n = 32$) was $\log10(x + 1)$ transformed.

| Trait | Field L | Field BR | Field DV | Field height | Field AL | Field ASR |
|---|---|---|---|---|---|---|
| **Lab L** | 0.350[*] | 0.169 | 0.110 | −0.102 | 0.027 | 0.202 |
| **Lab Location** | 0.189 | 0.094 | −0.147 | 0.044 | 0.156 | −0.272 |
| **Lab AL** | 0.144 | −0.079 | 0.067 | 0.227 | −0.385[*] | −0.090 |
| **Lab ASR** | −0.088 | −0.051 | −0.016 | 0.172 | −0.262 | −0.037 |

**Notes.**
L, latency; BR, Bite rate; DV, Distance ventured; H, Height; AL, Aggression latency; ASR, Aggression strike rate.
[*] Statistically significant at $p < 0.05$ level.

## DISCUSSION

Our study is one of the most detailed assessments of behavioral consistency of a marine organism to date. It shows that shortly after entering a new habitat at the end of their larval phase fish approximately three weeks old already have a complex repertoire of behaviors that are displayed in a consistent way through time, indicative of the existence of individual personalities. Moreover, this personality appears to be established prior to or immediately upon metamorphosis and settlement. Factors that are likely to favor consistent over conditional behavior, and thus give rise to individual personalities are diverse and include: genetic, physiological or developmental limits, costs of flexibility, costs and availability of information acquisition, metabolism, body size, or constraints on behavioral plasticity (*Sih, Bell & Johnson, 2004*; *Bergmuller, Schurch & Hamilton, 2010*; *Briffa & Greenaway, 2011*). Stable behavioral states are hypothesized to be created when positive feedback loops form between state variables such as size, competitive ability, or condition and state-dependent behavioral decisions (*Dall, Houston & McNamara, 2004*; *Sih & Bell, 2008*). For example, individuals with higher body condition may be more cooperative compared to those in poorer condition because they can afford the energy expenditure. If cooperative behavior then led to increased energy gains, this feedback loop would maintain higher body condition (*Bergmuller, Schurch & Hamilton, 2010*). Naïve juvenile reef fish exhibiting personalities at settlement suggests a genetic component and strong trade-offs related to adopting alternative personalities. High mortality rates at this phase of their life cycle could provide very strong selective force and are most likely to be involved (*McCormick & Meekan, 2010*).

Generally, our study found moderate to highly repeatable behavioral scores for almost all behavioral measures. These ranged from 0.33 (height on the habitat patch across multiple days) to 0.89 (height across camera observations), values well within the range recorded by earlier studies. A recent meta-analysis by *Bell, Hankison & Laskowski (2009)* reported an average repeatability value of 0.37 in various behavioral traits across 114 studies and 98 species. They found mating, habitat selection and aggression to be the most repeatable traits; while activity, mate preference, and migration were the least repeatable. Consistency was generally higher for behaviors measured at closer time intervals, juveniles compared to adults and field studies versus laboratory settings (*Bell, Hankison & Laskowski, 2009*). Approximately 70% of this distribution was between 0.1 and 0.6 (see Fig. 1, *Bell, Hankison & Laskowski, 2009*). An additional 11 studies published more recently (*Réale et al., 2000*; *Smith & Blumstein, 2008*; *Briffa & Greenaway, 2011*; *Marras et al., 2011*; *Couchoux & Cresswell, 2012*; *Carter et al., 2012*; *Beckmann & Biro, 2013*; *Neumann et al., 2013*; *Pruitt, Grinsted & Settepani, 2013*; *Kelley, Phillips & Evans, 2013*; *Burtka & Grindstaff, 2013*) reported repeatability scores ranging from as low as 0.14 for a measure of aggression in male crested macaques (*Macaca nigra*) (*Neumann et al., 2013*) to as high as 0.92 for a measure of escape response in European sea bass (*Dicentrarchus labrax*) (*Marras et al., 2011*). Despite the wide range in these scores, they were cited as evidence of the consistency of behaviors and therefore personalities. On this basis, the repeatability scores we obtained suggest evidence for personality in the 3-week old damselfish that were the subjects of our study.

Large confidence intervals around a repeatability estimate suggest significant within-individual variation in behavior (*Jones & Godin, 2009*). While juvenile damselfish are known to adopt a wide range of behavioral strategies (*White et al., 2013*; *White, McCormick & Meekan, 2013*), some of the variation we recorded may be due to plasticity in the amount of habituation to the experimental protocol (*Martin & Réale, 2008*). Across repeated trials, an environment or test may become less novel and individuals may habituate to novelty in itself (*Réale et al., 2007*; *Edwards et al., 2013*), or alternatively become less responsive or sensitized (*Budaev, 1997*; *Martin & Réale, 2008*; *Kelley, Phillips & Evans, 2013*). In our study, the tests that involved an experimental set-up, such as laboratory-based measurements of boldness (e.g., latency to emerge), have some of the largest confident intervals. However, given our significant repeatability estimates, we are confident all the measures reported are reliable measures of an individual's behavior within these contexts.

Variables that originated from the aggression assay (strike latency and mirror strike rate) were the only measurements found not to be repeatable through time or context. This suggests the moderate negative correlation found between field and laboratory measures of aggression strike latency is likely to be ecologically irrelevant. While a commonly-used test (*Gerlai, 2003*; *Marks et al., 2005*), these measures may be susceptible to the habituation effect discussed above. A closely-related species, *P. moluccensis*, has been shown to recognize threats after a single exposure (*Mitchell et al., 2011*). Perhaps *P. amboinensis* similarly learns to ignore the false threat of their reflection after repeated exposures.

Observations repeated over short time scales (4 min apart, simultaneous video vs. video observations) had the highest repeatability scores. Measures conducted over longer (30 min apart and 3 times daily over 3 days) time periods had similar, but lower scores. This agrees with results from a meta-analysis, which showed higher estimates of repeatability for behaviors measured at shorter time intervals (*Bell, Hankison & Laskowski, 2009*). Our results suggest juvenile damselfish quickly adopt stable behavioral phenotypes regarding foraging and activity rates following settlement and remain consistent throughout the intense predation pressure experienced during the first few days on the reef.

There was a trend for repeatability estimates obtained in the laboratory to be lower compared to field-based measurements. This same pattern was found in *Bell, Hankison & Laskowski*'s (*2009*) meta-analysis. If there are advantages to behaving consistently (*Dall, Houston & McNamara, 2004*; *McElreath & Strimling, 2006*), then the greater environmental variance in the field might create micro-niches, increasing repeatability by allowing individual expression of behavioral variations (*Bell, Hankison & Laskowski, 2009*). Also, because juveniles are exposed to innately higher predation pressure in the field, this could act as a directional or stabilizing selection on behavior (*Bell, Hankison & Laskowski, 2009*). However, in this study fish are initially naïve and neophobic upon introduction to the field (*Meekan et al., 2010*; *Chivers et al., 2014*; *Ferrari et al., 2015*), so perhaps the greater sensory input in the field environment is enough to act as a stabilizing influence. A recent study found three-spined sticklebacks (*Gasterosteus aculeatus*) adopted stable boldness-aggressiveness correlations once exposed to predators (*Bell & Sih, 2007*). Juvenile damselfish quickly learn about predators (*Mitchell et al., 2011*) and are likely to swiftly

adopt a consistent behavioral phenotype when faced with the variations and challenges of their natural habitat. Given the few and weak correlations found between field and laboratory measures, and lower consistency for laboratory studies suggests inferences about natural behaviors in the field derived from laboratory studies need to be made cautiously (*White, McCormick & Meekan, 2013*). The lack of predators and increased novelty of the laboratory environment may enable juvenile damselfish to exhibit a great variability of behaviors or prompt different behavioral responses that have little bearing on likely behavior under natural conditions. This implies behavioral studies have limited predictive ability across situations, in particular using laboratory measures to predict behaviors in the field (*White et al., 2013*; *White, McCormick & Meekan, 2013*).

Interestingly, *Beckmann & Biro (2013)* reported repeatability values almost identical to ours for the same laboratory-based boldness measure. They tested two species of juvenile damselfish (*P. wardi* and *P. amboinensis*) and showed repeatability in the emergence latency test in home tanks ($R = 0.42$ for *P. amboinensis* on the third observation), but no correlations when compared against the same and different behavioral tests in different contexts. Others have also argued for the use of multiple measures of boldness in order to obtain an ecologically relevant assessment of this behavioral trait (*White et al., 2013*), and have also found a lack of behavioral consistency across situations (*White, McCormick & Meekan, 2013*) for juvenile damselfish. While *Beckmann & Biro (2013)* argue the lack of correlation across contexts means this assay is inadequate to measure boldness, their study likely had issues with habituation (*Edwards et al., 2013*). In contrast, we found latency to emerge behavior to be significantly repeatable within a single context and moderately positively correlated with an emergence test in the field.

Another important result of our study was that the presence of observers seemed to have no significant impact on fish behavior. While fishes are the focus of much behavioral research, they are rarely observed in their natural environments (*Réale et al., 2000*; *Bell, Hankison & Laskowski, 2009*). Typically, observations in a field situation would be conducted from behind a blind (*Martin & Bateson, 2007*), a luxury not afforded to a noisy bubble-blowing SCUBA diver. While the simultaneous observer and video observations had slightly lower repeatability scores for bite rate and height compared to the comparison of the two video scores (difference of 0.23 and 0.13, respectively), this is most likely an artifact of the difficulties associated with observing detailed behavior via camera. Even with high resolution video, it was difficult to distinguish between feeding strikes and the natural stop–start swimming of these fish. Also, fish leaving the field of view of the camera for a short duration was not an issue for the diver who could maintain visual contact with the target fish at all times. Overall, discrepancies between the methods of observation may have resulted in a slight over-counting of bite rates in the video. This suggests video data is less useful for subjects such as these small damselfish that are quick moving and very mobile. As long as slow, deliberate movements are employed and the observer remains a least a meter away, juvenile damselfish seem indifferent to human presence thus diver observations provide useful records of behavior.

In summary, our results demonstrate that measures of boldness and activity, both in the field and the laboratory, are highly repeatable over time scales relevant to this species during a key period of their life history. These stable behaviors indicate that these 3-week old juvenile fish already have personalities. From a methodological perspective, our results indicate that an initial 3 min assessment of their behavior provides a useful record of an individual's personality. However, caution is required when comparing field and laboratory based behaviors (*White et al., 2013*). Future studies with this species can reasonably use a single (i.e., unrepeated) assay to reduce animal stress, which can then be correlated with physical measures of performance and success to determine how individual characteristics combine to affect fitness. Future research will investigate if adult *P. amboinensis* retain this behavioral consistency through ontogeny.

## ACKNOWLEDGEMENTS

We would like to thank C Mero for collecting some of the initial data associated with the project. S Banana provided guidance and support throughout the study. J Smart kindly assisted in the field and we thank the staff of Lizard Island Research Station (Australia Museum) for generous logistical support.

### Funding

This research was funded by an ARC Discovery grant (CE140100020) to Mark I. McCormick and through the ARC Centre of Excellence for Coral Reef Studies (DP120101993). The funders had no role in study design, data collection and analysis, decision to publish, or preparation of the manuscript.

### Grant Disclosures

The following grant information was disclosed by the authors:
ARC Discovery: CE140100020.
ARC Centre of Excellence for Coral Reef Studies: DP120101993.

### Competing Interests

The authors declare there are no competing interests.

### Author Contributions

- James R. White conceived and designed the experiments, performed the experiments, analyzed the data, wrote the paper, prepared figures and/or tables.
- Mark G. Meekan and Mark I. McCormick conceived and designed the experiments, reviewed drafts of the paper.

### Animal Ethics

The following information was supplied relating to ethical approvals (i.e., approving body and any reference numbers):

Fish collection locations/activities and handling protocols were approved by the Great Barrier Reef Marine Park Authority (Permit Number: G10/33784.1) and JCU Animal Ethics Committee (Permit Number: A1067).

### Field Study Permissions

The following information was supplied relating to field study approvals (i.e., approving body and any reference numbers):

Fish collection locations/activities and handling protocols were approved by the Great Barrier Reef Marine Park Authority (Permit Number: G10/33784.1) and JCU Animal Ethics Committee (Permit Number: A1067).

### Supplemental Information

Supplemental information for this article can be found online at http://dx.doi.org/ 10.7717/peerj.961#supplemental-information.

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
