# Peer review of "Individual consistency in the behaviors of newly-settled reef fish"

_PeerJ, doi:10.7717/peerj.961_

## Round 0.1 · original submission · Minor Revisions

Reviewer 1 in particular has some helpful suggestions that I would like to see incorporated into the revision.

Also, it seems to me that one of the more interesting findings is the lack of correspondence between field and lab behavior. It seems to me that this difference could be played up more in the introduction and the discussion.

Also please attend to the following grammatical issues:

Please replace “since’ with “because” and “while” with "whereas" when not referring to temporal changes (for example lines 60 and 63).

On lines 196 and 231 change “was” to “were”.

·

Basic reporting

Altogether I found the article to be well written and for the most part, easy to understand. I had a few confusions about some of the methods - it took me a few reads of the methods to figure out exactly which fish were used for what. I find the methodology itself to be fine, but I think some tweaking of how you present your methods will improve the understanding of future readers.

Paragraph line 114: At first I had trouble figuring out that you collected different cohorts of fish at different times for each of your experiments. I think this is because you presented the final sample size right at the beginning of this paragraph which made me think 110 fish were all collected at the same time. I think rather it be helpful to start this paragraph by explicitly stating that different cohorts of fish were used for different experiments (right now this is stated for the first time on line 128), but that all of them were housed in captivity first (i.e. I think the fish you mention on Line 129-130 are part of the fish mentioned on line 114?). Then within each description of each experiment (short term consistency in the field, short term in the lab, etc) you can mention the sample size and year collected and then include a total sample size at the very end of the methods.

I would also suggest that you move the observer vs video comparison to the end of the description of the field methods. I think it would make more sense to first describe the behaviors you measured and then as a sidenote, mention that you also measured these behaviors using observer and a camera (this comparision isn't really the main focus of your paper so maybe doesn't deserve to be so prominent). And just as a small detail of personal preference, I didn't find having the letters "A", "B", "C" associated with observer, simultaneous video and video observations were helpful - they actually threw me for a loop when I saw them in the table! (it's probably just easier to call them "observer vs. video" "video vs video" or something like that).

Field vs lab - If there are no space restrictions then there's no reason not to include the correlation matrix for the field and lab behaviors (if there are space restrictions then these can be thrown into a supplemental). It's just reassuring to the reader to see the results themselves (as opposed to being told they are just not signficant).

Finally, I think it's important for the reader to just get an idea for how the animal behaved in general and I think the paper would really benefit from a figure showing the individual reaction norm graphs (time on the x axis, behavior on the y, each individual as a different line). This would be helpful for two reasons – first, and perhaps most importantly, it would assure the reader that your individuals actually did show an appropriate level of variation in behavior (e.g. not all individuals had a very short or very long (maximum) latency time to emerge from the shelter) and second, it would really drive home the point that individuals do consistently differ in behavior. Perhaps this doesn’t need to be done for all behaviors, but assuming there aren’t major space restrictions, I think it would be helpful to see this for at least one behavior in each condition (field, lab).

Small detail - In Table 1 you should switch around the order of the CI so the lower bound is first.

Experimental design

No major concerns, just a few details to add:

Line 120 - You say each fish was acclimitized for "at least 24 hours" prior to experimentation, does this mean all fish from all cohorts (experiments) were kept in captivity for ONLY 24 hours or did this differ among experiments? This could be clarified within each description of the each experiment.

Observer vs. video - which cohort of fish was this performed on? It was unclear to me whether this was done on the 18 fish that you assessed for short term consistency or the 21 fish you observed over multiple days? I think just moving this part to after the behavioral description (as mentioned in my comments above) would fix this problem.

Data analysis - did you include any other fixed effects? Given that you have body size of each individual and body size is known to influence behavior, it would be a good idea to include this in your models to see if differences in body size are driving these individual differences in behavior (or not).

Validity of the findings

No comments.

Reviewer 2 ·

Basic reporting

No Comments

Experimental design

No Comments

Validity of the findings

No Comments

Additional comments

This is an interesting study of consistency in several aspects of the behaviour in a reef fish. The study of personality has become a mainstream in ethology and behavioural ecology and the fact that individuals are consistent over time and across contexts is now far from surprising. Indeed, consistency is what is currently expected. On the other hand, there is still a scarcity of data on behavioural consistency early in ontogeny. It is not always clear how early consistency appear, for example. Thus, even though the paper is far from new in its subject or perspective, it does present interesting and novel data on early appearance of consistent individuality. Additionally, the data comparing consistency in of the fish the lab and in the field is also new and methodologically interesting. Thus, I recommend accepting this manuscript with small revisions.

Comments:

lns. 48-50. I found the discussion of the three forms of consistency confusing. It seems that some of them are perhaps not consistency of individual differences at all. Probably it is better to drastically abridge this passage, it is also not crucial for the study. In my opinion it is enough to define what is consistency, note plasticity, consistency over time and contexts.

Lns. 152-153. There is clearly an issue of habituation to the context, the as two observation periods are always in the same order. A little discussion might be necessary.

Lns. 182. This figure is unavailable to the reader, remove from text. If absolute necessary, we need a figure here.

Lns. 303. The “medium” interval = 30 min should actually be considered rather short.

---

## Round 0.2 · Minor Revisions

Thank you for being responsive to the reviewers' suggestions in your revision. I notice some very minor things that should be changed before the paper can be formally accepted.

1. You still refer to short, medium and longer time scales on lines 167-168.

2. You indicate that "different cohorts of fish were used for the different consistency experiments" (lines 211-213). This sentence is confusing. How can you measure consistency with different individuals? This confusion also occurs on lines 226-227. Perhaps describe as just different experiments, omitting the word "consistency" such that it is clear that the behavior of all fish were compared across conditions to measure consistency but that different intervals and measures were taken with different groups of fish. I realize it is probably very clear to you but it still reads as somewhat confusing for the reader.

3. Given that there were few correlations between field and lab studies, and that you found generally lower consistency for lab studies, I would still like to see a greater emphasis on these potential limitations of lab studies in the discussion.

4. On line 412, please insert "that" before "the presence of observers".

---

## Round 0.3 · accepted · Accept

Thank you very much for your prompt attention to the last few requested changes. Thank you for submitting a nice piece of work to Pee J. I am now pleased to offer formal acceptance of your manuscript.